# CAN PRE-TRAINED NETWORKS DETECT FAMILIAR OUT-OF-DISTRIBUTION DATA?

## ABSTRACT

Out-of-distribution (OOD) detection is critical for safety-sensitive machine learning applications and has been extensively studied, yielding a plethora of methods developed in the literature. However, most studies for OOD detection did not use pre-trained models and trained a backbone from scratch (Yang et al., 2022). In recent years, transferring knowledge from large pre-trained models to downstream tasks by lightweight tuning has become mainstream for training in-distribution (ID) classifiers. To bridge the gap between the practice of OOD detection and current classifiers, the unique and crucial problem is that the samples whose information networks know often come as OOD input. We consider that such data may significantly affect the performance of large pre-trained networks because the discriminability of these OOD data depends on the pre-training algorithm. Here, we define such OOD data as PT-OOD (**P**re-**T**rained **OOD**) data. In this paper, we aim to reveal the effect of PT-OOD on the OOD detection performance of pre-trained networks from the perspective of pre-training algorithms. To achieve this, we explore the PT-OOD detection performance of supervised and self-supervised pre-training algorithms with linear-probing tuning, the most common efficient tuning method. Through our experiments and analysis, we find that the low linear separability of PT-OOD in the feature space heavily degrades the PT-OOD detection performance, and self-supervised models are more vulnerable to PT-OOD than supervised pre-trained models, even with state-of-the-art detection methods. To solve this vulnerability, we further propose a unique solution to large-scale pre-trained models: Leveraging powerful instance-by-instance discriminative representations of pre-trained models and detecting OOD in the feature space independent of the ID decision boundaries. This study provides significant insights to OOD detection with pre-trained models.

## 1 INTRODUCTION

Deep neural networks perform successfully when the testing data is drawn from the same distribution as the training data. This kind of test-time data with distributions matching the training-time data is called in-distribution (ID) data. However, out-of-distribution (OOD) samples that do not belong to ID categories are likely to occur in the testing data. When deep learning is deployed in many applications, such as autonomous driving (Blum et al., 2019), medical applications (Ulmer et al., 2020), and web applications (Aizawa & Ogawa, 2015), it becomes crucial to detect OOD samples.

Recently, the development of large-scale pre-trained models (Chen et al., 2020a;b; Dosovitskiy et al., 2021; Caron et al.,

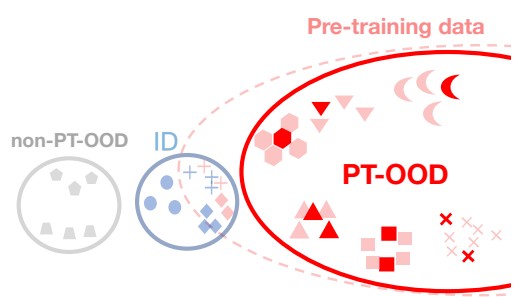

Figure 1: **Relationship between large pre-training data, in-distribution (ID) data, and PT-OOD data.** A large number of data similar to the large pre-training data and not in the ID classes can be PT-OOD data.

2021; Wightman et al., 2021) has made it common to retain the knowledge of these models and use their rich representations for downstream tasks (Islam et al., 2021; Kotar et al., 2021; Vasconcelos et al., 2022; Yamada & Otani, 2022). In particular, leveraging large pre-trained knowledge and tuning it to downstream tasks in a lightweight manner without destroying the knowledge is an important challenge today for real-world applications (Houlsby et al., 2019; Jia et al., 2022; Hu et al., 2022). We consider that OOD detection is no exception when it comes to transferring pre-trained knowledge. Most existing OOD detection research (Yang et al., 2022; Zhang et al., 2023) has not used large pre-trained models for training the ID classifier to fairly compare the proposed methods with the conventional baseline methods (Hendrycks & Gimpel, 2017; Liang et al., 2018; Lee et al., 2018; Liu et al., 2020; Huang et al., 2021; Sun et al., 2022; Wang et al., 2022; Ahn et al., 2023). However, as the number of classifiers based on pre-trained models increases, it becomes essential to investigate the robustness of OOD detection for these classifiers to ensure their safety.

In previous settings without large-scale pre-trained models, OOD detection aims to detect completely novel OOD samples that networks do not know. However, we consider that this motivation does not always hold true today when many ID classifiers are trained using large-scale pre-trained models. For example, currently, numerous images scraped from the web are used as pre-training data (Deng et al., 2009; Ridnik et al., 2021; Radford et al., 2021). If the downstream deep learning application is deployed on the web, malicious users might intentionally input OOD images on the web (*i.e.*, pre-training data) to applications. As the size of pre-training data increases, this problem occurs more commonly. Therefore, there is a growing demand for research on OOD detection in the setting where there exists an overlap between pre-training data and OOD data.

Based on the above current situation where (i) lightweight tuning with keeping the pre-trained knowledge has become mainstream and (ii) data similar to the pre-training data comes in as OOD, we tackle the novel question: "Can pre-trained networks correctly identify OOD samples whose information the backbone networks have?". We refer to such OOD samples with the overlap of pre-training data as "*PT-OOD*" (**P**re-**T**rained **OOD**) data. The relationship between ID, non-PT-OOD, and PT-OOD is shown in Fig. 1. The data that are similar to the pre-training data and not in the ID classes can be PT-OOD data. The goal of this task is to detect PT-OOD at inference, *i.e.*, to perform a binary classification of whether the test data is ID or OOD for test-time PT-OOD data (a kind of OOD data) (see Fig. 3 in Appendix A for the illustration). To investigate the robustness to PT-OOD, we focus on pre-training algorithms. That is because the discriminability of PT-OOD data depends on the pre-training algorithm, and we consider the difference in discriminability to influence the detection performance. In particular, we investigate the OOD detection robustness for supervised pre-training (Wightman et al., 2021; Touvron et al., 2021) and self-supervised pre-training (Grill et al., 2020; Caron et al., 2020; Chen et al., 2020b; Caron et al., 2021) with both CNN (He et al., 2016) and Transformer (Dosovitskiy et al., 2021) architectures. For lightweight tuning methods, we adopt a linear probing strategy (updating only the last linear layer) because it is the most common lightweight tuning method.

Through comprehensive experiments, we find that the PT-OOD detection performance differs significantly among pre-training methods. In particular, supervised pre-trained models can correctly detect PT-OOD samples, but self-supervised pre-trained models have a lower PT-OOD detection performance. This result holds true even when self-supervised methods outperform supervised methods in ID accuracy or when state-of-the-art OOD detectors (Liu et al., 2020; Huang et al., 2021; Sun et al., 2021; Hendrycks et al., 2022) are used. This finding is not trivial because existing work states that high ID classification performance contributes to high OOD detection performance (Vaze et al., 2022). Our analysis attributes the detection performance of PT-OOD to the linear separability of PT-OOD in the pre-trained feature space. By pushing the samples in the same class into the same embeddings, supervised pre-training has a high linear separability on PT-OOD in the feature space. Therefore, PT-OOD data are not scattered but concentrated in the feature space, so they do not come inside the ID decision boundary. In contrast, self-supervised pre-training aims to learn transferable representations. Hence, PT-OOD are scattered in the feature space and they might enter inside the ID decision boundary. This analysis holds true for recent foundation models (*e.g.,* CLIP), and reveals that CLIP has a vulnerability to PT-OOD. The transferability of pre-training algorithms is known to be beneficial for improving ID classification accuracy (Islam et al., 2021), but we find that it may cause vulnerability in OOD detection, which is a remarkable finding in this paper.

To detect PT-OOD with various pre-trained models, we further propose a solution unique to the use of large-scale pre-trained models. Without pre-trained models, we need to utilize ID decision

boundaries to separate ID from OOD for any detection method. However, when leveraging large-scale pre-trained models, we can utilize instance-by-instance discriminative features to separate ID and OOD, which require the ID decision boundaries. Therefore, by applying feature-based OOD detection methods, *e.g.*, kNN (Sun et al., 2022), to large pre-trained models *directly* (*i.e.*, without using ID decision boundaries), we can detect PT-OOD samples with various pre-trained models. The contributions of our paper are summarized as follows:

- We tackle a novel problem PT-OOD detection from the perspective of pre-training algorithms, which is unique and crucial for exploring the OOD detection performance of current classifiers with large pre-trained knowledge (see Fig. 1 and 3).
- From the experimental results and discussion, we find that the linear separability on PT-OOD is attributed to the performance of OOD detection, and self-supervised models have lower detection performance than supervised models (see Table 1, 2, 3, and Fig. 2).
- To detect PT-OOD samples with various pre-training methods, we propose to leverage the instance-by-instance features of large pre-trained models without using ID decision boundaries (see Table 4).

## 2 RELATED WORK

**OOD detection with pre-trained models.** OOD detection has long been an active research area in the context of the safety of machine learning applications, and numerous detection methods have been proposed in the literature (Yang et al., 2022; Zhang et al., 2023). Most OOD detection work (Liang et al., 2018; Lee et al., 2018; Liu et al., 2020; Huang et al., 2021; Wang et al., 2022; Sun et al., 2022) assumes training ID classifiers from scratch to fairly compare the proposed methods with the conventional baseline methods (Hendrycks & Gimpel, 2017). However, these days, the mainstream of training ID classifiers is to leverage the powerful representations of pre-trained models and tune them in a lightweight way (*e.g.*, linear-probing). This lightweight strategy is prevalent, especially in common real-world situations where computational resources and data are limited. Therefore, there is a gap between the ID classifier used in existing OOD detection research and the ID classifier used in real-world applications. To avoid misunderstanding, we would like to note that there are some existing papers that use pre-trained models for OOD detection. For example, Hendrycks et al. (2019a) explored the OOD detection robustness of supervised pre-trained models and stated that using pre-trained models improves the OOD detection performance. However, we consider that this conclusion cannot always be generalized because (i) they used only supervised pre-training methods and did not investigate the robustness of other pre-training algorithms (*e.g.*, self-supervised methods), (ii) they conducted experiments on only small-scaled datasets for both pre-training and fine-tuning and did not investigate the robustness with real-world datasets (*e.g.*, ImageNet) and (iii) they performed fully fine-tuning and did not investigate the robustness of lightweight tuning methods. Similarly, some subsequent work using pre-trained models (Koner et al., 2021; Fort et al., 2021) performed full-tuning and also used the OOD data (*e.g.*, low-resolution images) completely different from the pre-trained data, so the effect of pre-training on OOD detection has not been fully investigated. Hence, this is the first study to investigate the OOD detection performance of various pre-training approaches when transferring large pre-trained knowledge.

**OOD dataset in previous work.** Existing OOD detection benchmarks are chosen based on whether ID and OOD are semantically near (near OOD) or far (far OOD) (Yang et al., 2022; 2021; Yu & Aizawa, 2019; Winkens et al., 2020; Zhang et al., 2023). This is because whether ID data is close or far affects the difficulty of OOD detection. In addition to this perspective, when transferring large pre-trained knowledge to the ID classifier, we consider whether OOD data is close to pre-training data is also an important factor in affecting the OOD detection performance. This is because different pre-training strategies learn different representations of PT-OOD, and we consider the difference in the discriminability of PT-OOD to influence the detection performance. Also, since a large amount of pre-training data is scraped from the web, it is more likely for PT-OOD to come in as input. This paper explores how PT-OOD data can affect detection performance with various pre-training algorithms.

**Robustness of supervised and self-supervised pre-training.** The robustness of supervised and self-supervised pre-training has been explored in many fields, such as the robustness for dataset imbalance setting (Liu et al., 2022), OOD/domain generalization (Zhong et al., 2022; Kim et al.,

2022; Huang et al., 2023). However, since the robust pre-training algorithm varies from task to task (Liu et al., 2022; Kim et al., 2022), the robustness for other tasks does not hold for OOD detection. Hence, it is necessary to explore OOD detection robustness. Some existing methods, such as Rot (Hendrycks et al., 2019b), CSI (Tack et al., 2020), and SSD (Sehwag et al., 2021), incorporate self-supervised training for OOD detection, but they use self-supervised training at the ID training stage, not the pre-training stage. Therefore, this is the first work to explore the robustness when transferring the knowledge of various pre-trained models for OOD detection.

## 3 ANALYSIS SETUP

### 3.1 PROBLEM SETTING

For OOD detection with pre-trained models (Hendrycks et al., 2019a; Fort et al., 2021; Li et al., 2023), the ID classes refer to the classes used in the downstream classification task, which are different from the classes of the upstream pre-training. OOD classes are the classes that do not belong to any of the ID classes of the downstream task. Our setting considers the overlap between pre-training and OOD data, as well as the discrepancy in the semantics between ID and OOD. In this study, we define PT-OOD (a kind of OOD data) as having high visual (domain) and semantic (class) similarity as pre-training data. We study the scenario where the model is pre-trained with a large-scale dataset and then fine-tuned on the training dataset without any access to OOD data. During testing, we detect PT-OOD. The illustration for our setting is shown in Fig. 3 in Appendix A.

### 3.2 DATASET

**Pre-training dataset.** We use ImageNet-1K (Deng et al., 2009) as the pre-training data. ImageNet-1K contains 1.2M images of mutually exclusive 1,000 classes. Although ImageNet-22K (Ridnik et al., 2021) and JFT-300M (Sun et al., 2017) are larger as pre-training data than ImageNet-1K, most self-supervised pre-trained models are not publicly available for these datasets. Therefore, using ImageNet-1K pre-trained models best suits this study's purpose, which is to make fair comparisons with different pre-training algorithms. Note that for the fair comparisons and limitation of publicly available models, the main experiments are conducted on ImageNet-1K, but our findings and analysis are applicable to the recent more large-scale pre-trained model CLIP (Radford et al., 2021) (refer to Sec. 5).

**ID dataset.** Following the existing literature (Liang et al., 2018; Koner et al., 2021; Fort et al., 2021; Ming et al., 2022b), we use CIFAR-10 (Krizhevsky, 2009) and CIFAR-100 (Krizhevsky, 2009) as ID datasets in the main experiments. CIFAR-10 consists of 50,000 training images and 10,000 test images with 10 classes. CIFAR-100 consists of 50,000 training images and 10,000 test images with 100 classes. To conduct a comprehensive study, we also use Caltech-101 (Nilsback & Zisserman, 2008) and Food-101 (Bossard et al., 2014) as ID datasets. Caltech-101 consists of pictures of objects belonging to 101 classes. Food-101 contains the 101 most popular and consistently named dishes. For Caltech-101, we exclude two face-relevant classes because OOD images might have some persons who are relevant to face classes. For all datasets, we use a size of 224×224 following previous work (Koner et al., 2021).

**PT-OOD dataset.** As PT-OOD datasets, we use two ImageNet subsets datasets. First is ImageNet-30 test data (IN-30) (Hendrycks et al., 2019b), which consists of 30 classes selected from validation data and an expanded test data of ImageNet-1K. We use ImageNet-30 as PT-OOD except for Caltech. The second is ImageNet-20-O (IN-20) where we collected samples in 20 classes from ImageNet validation data. To avoid the overlap of classes with ID data, we exclude some classes and samples for each ID dataset. To assume a realistic setting, we use the test and validation set of ImageNet, which are not pre-training data itself but have high visual and semantical similarity to the pre-training data. We resize the images in ImageNet-30 and ImageNet-20-O to a size of 224×224. Detailed information is shown in Appendix C.

**Non-PT-OOD dataset.** Although we have primarily focused on PT-OOD whose information the pre-trained models have, we also need to consider non-PT-OOD whose information they do not have (*i.e.*, completely novel OOD data for networks). As non-PT-OOD datasets, we use Resized LSUN (LSUN-R) (Liang et al., 2018), iSUN (Xu et al., 2015), CIFAR-100 (when ID is CIFAR-10), and CIFAR-10 (when ID is CIFAR-100) following the existing studies with pre-trained models (Fort

Table 1: **Results for the OOD detection robustness of pre-trained models.** MSP (Hendrycks & Gimpel, 2017) is used as a detection method. LSUN-R/iSUN/CIFAR-100 (CIFAR-10) are non-PT-OOD, and IN-30 and IN-20 are PT-OOD. Supervised learning with † denotes the baseline. Gaps of 5% or less are shown in gray, 5% to 10% in black, and 10% or more in red. The result shows that supervised methods have significantly higher PT-OOD detection performance than self-supervised methods.

(a) CIFAR-10

| Backbone | Method | ID acc. (%) | AUROC (%) | | | | |
|---|---|---|---|---|---|---|---|
| | | | LSUN-R | iSUN | CIFAR-100 | IN-30 | IN-20 |
| ResNet-50 | Sup.† | 90.73 | **90.34** | **88.50** | **88.31** | **95.77** | **94.96** |
| | BYOL | **92.15**↑+1.42 | 88.40↓-1.94 | 87.57↓-0.93 | 86.64↓-1.67 | 86.02↓-9.75 | 86.99↓-7.97 |
| | SwAV | 91.93↑+1.20 | 86.52↓-3.82 | 86.47↓-2.03 | 84.98↓-3.33 | 81.05↓-14.72 | 83.94↓-11.02 |
| | MoCo v2 | 90.42↓-0.31 | 87.57↓-2.77 | 87.20↓-1.30 | 84.84↓-3.47 | 69.39↓-26.38 | 73.39↓-21.57 |
| ViT-S | Sup.† | 92.52 | **93.07** | **92.43** | **90.67** | **98.16** | **97.76** |
| | iBOT | **93.75**↑+1.23 | 90.05↓-3.02 | 90.03↓-2.40 | 87.49↓-3.18 | 80.68↓-17.48 | 86.09↓-11.67 |
| | DINO | 92.29↓-0.23 | 86.89↓-6.18 | 86.60↓-5.83 | 86.11↓-4.56 | 80.00↓-18.16 | 83.93↓-13.83 |

(b) CIFAR-100

| Backbone | Method | ID acc. (%) | AUROC (%) | | | | |
|---|---|---|---|---|---|---|---|
| | | | LSUN-R | iSUN | CIFAR-10 | IN-30 | IN-20 |
| ResNet-50 | Sup.† | 73.77 | 74.14 | 73.77 | **73.88** | **79.98** | **81.42** |
| | BYOL | **76.54**↑+2.77 | **75.12**↑+0.98 | 74.14↑+0.37 | 72.87↓-1.01 | 72.50↓-7.48 | 76.72↓-4.70 |
| | SwAV | 76.33↑+2.56 | 74.47↑+0.33 | 74.22↑+0.45 | 73.28↓-0.60 | 71.67↓-8.31 | 71.72↓-9.70 |
| | MoCo v2 | 71.91↓-1.86 | 68.90↓-5.24 | 68.01↓-5.76 | 69.22↓-4.66 | 49.23↓-30.75 | 53.85↓-27.57 |
| ViT-S | Sup.† | 75.73 | 73.90 | 75.20 | **75.68** | **89.54** | **90.78** |
| | iBOT | **78.45**↑+2.72 | **79.66**↑+5.76 | **78.83**↑+3.63 | 74.42↓-1.26 | 55.51↓-34.03 | 62.42↓-28.36 |
| | DINO | 76.49↑+0.76 | 78.19↑+4.29 | 78.24↑+3.04 | 73.29↓-2.39 | 56.62↓-32.92 | 63.97↓-26.81 |

(c) Caltech-101

| Method | ID acc. (%) | AUROC (%) | |
|---|---|---|---|
| | | iSUN | IN-20 |
| Sup.† | 95.20 | **99.06** | **97.52** |
| BYOL | **95.40**↑+0.20 | 97.88↓-1.18 | 94.83↓-2.69 |
| SwAV | 95.00↓-0.20 | 98.46↓-0.60 | 92.12↓-5.40 |
| MoCo v2 | 93.10↓-2.10 | 96.35↓-2.71 | 85.36↓-12.16 |

(d) Food

| Method | ID acc. (%) | AUROC (%) | |
|---|---|---|---|
| | | iSUN | IN-30 |
| Sup.† | 69.66 | **97.93** | **93.57** |
| BYOL | **73.70**↑+4.04 | 95.13↓-2.80 | 86.96↓-6.61 |
| SwAV | 72.99↑+3.33 | 90.75↓-7.18 | 83.12↓-10.45 |
| MoCo v2 | 68.27↓-1.39 | 90.20↓-7.73 | 68.66↓-24.91 |

et al., 2021; Koner et al., 2021). These datasets are low-resolution images, and the domain is different from the pre-training data, so the pre-trained network would not have information on such images. We use a size of 224×224 following previous work (Koner et al., 2021). For more detailed experiments, we also used datasets such as iNaturalist (Van Horn et al., 2018) and SUN (Xiao et al., 2010), which are visually similar to ImageNet but have no overlap in semantics. We included these results in Appendix B.5.

### 3.3 PRE-TRAINED MODELS

As the backbone network, we use ResNet-50 (He et al., 2016) and Vision Transformer Small (ViT-S) (which follows the design of DeiT-S (Touvron et al., 2021)). The choice of ViT-S is motivated by its similarity with the number of parameters (21M for ResNet-50 and 23M for ViT-S).

**Supervised learning.** For supervised methods, we use the supervised models trained with the latest training recipe, such as data augmentation (Zhang et al., 2018; Yun et al., 2019; Cubuk et al., 2020), or optimizers (Kingma & Ba, 2015; You et al., 2020). In particular, for ResNet-50, we use the pre-trained model provided by timm library (Wightman et al., 2021). For ViT-S, we use the pre-trained models provided by DeiT (Touvron et al., 2021). These models are both pre-trained with

Table 2: **Results with the state-of-the-art detection methods and MLP** on PT-OOD. We use CIFAR-10 and CIFAR-100 as ID, and ImageNet-30 as OOD.

| ID | Method | MSP | Energy | ReAct | MaxLogit | GradNorm | DICE | MLP |
|---|---|---|---|---|---|---|---|---|
| | Sup. | **95.77** | **98.90** | **99.16** | **98.60** | **93.31** | **99.74** | **84.81** |
| CIFAR-10 | SwAV | 81.05 | 84.88 | 86.97 | 84.99 | 69.44 | 98.60 | 79.36 |
| | MoCo v2 | 69.39 | 62.16 | 66.31 | 62.95 | 31.81 | 86.80 | 73.22 |
| | Sup. | **79.98** | **90.36** | **91.21** | **89.11** | **82.45** | **98.67** | **72.22** |
| CIFAR-100 | SwAV | 71.67 | 73.82 | 77.16 | 74.08 | 64.80 | 96.70 | 71.96 |
| | MoCo v2 | 49.23 | 34.94 | 41.02 | 35.80 | 17.24 | 63.65 | 49.10 |

ImageNet-1K.

**Self-supervised learning.** For self-supervised methods, we use BYOL (Grill et al., 2020), SwAV (Caron et al., 2020) and MoCo v2 (Chen et al., 2020b) for ResNet-50 and iBOT (Zhou et al., 2022) and DINO (Caron et al., 2021) for ViT. For ViT, Masked AutoEncoder (MAE) (He et al., 2022) is well known for self-supervised learning. However, MAE has a significantly low ID classification accuracy with linear probing (He et al., 2022; Xie et al., 2022) because it reconstructs raw pixels, which means that the representation of the last stage contains more low-level information and is not suitable for classification without fine-tuning. Therefore, we did not use MAE and its variants (He et al., 2022; Xie et al., 2022; Huang et al., 2022) for comparisons.

## 3.4 TRANSFER LEARNING

The goal of this study is to explore the OOD detection performance with current classifiers, where pre-trained knowledge is transferred in a lightweight way. Although many lightweight tuning methods have been proposed these days (Houlsby et al., 2019; Jia et al., 2022; Hu et al., 2022), we use a linear-probing strategy, which updates only the last linear layer—the head. This is because linear-probing is most widely used for lightweight tuning methods, and it is crucial to examine the OOD detection robustness of the most common tuning strategy.

## 3.5 EVALUATION METRICS

For evaluation, we use the area under the receiver operating characteristic curve (AUROC) (Davis & Goadrich, 2006). A perfect detector corresponds to an AUROC score of 100%, and a random detector corresponds to an AUROC score of 50%.

## 4 ROBUSTNESS OF PRE-TRAINED MODELS

**Main results.** In this section, we evaluate the OOD detection robustness of pre-trained models. For a detection method, we first use MSP (Hendrycks & Gimpel, 2017), which uses the ID classifier's maximum softmax probability. MSP is commonly used to compare OOD detection robustness between different models (Hendrycks et al., 2020; Koner et al., 2021; Fort et al., 2021). In Table 1, we show the OOD detection results with various pre-training methods on CIFAR-10, CIFAR-100, Caltech-101, and Food-101. We define the supervised method as the baseline for both ResNet and ViT, and gaps from the supervised method are shown. We find that in most settings, ID accuracy and non-PT-OOD detection performance are comparable, but PT-OOD detection performance varies greatly depending on the pre-training methods. In particular, the supervised methods have high PT-OOD detection abilities, while self-supervised learning methods have low PT-OOD detection abilities. For example, MoCo v2 has over 20.0 points lower than the supervised method on PT-OOD (IN-30, IN-20), although MoCo v2 has almost the same performance in ID accuracy and non-PT-OOD detection. Especially when ID is CIFAR-100, the AUROC score on PT-OOD is around 50.0%, similar to that of a random OOD detector (AUROC is 50.0). For BYOL and SwAV, although they outperform the supervised method in ID accuracy, the PT-OOD detection performance tends to be lower than the supervised method in most settings. Similarly in ViT, iBOT and DINO have significantly lower PT-OOD detection performance than the supervised pre-training method. These results in Table 1 indicate that the supervised methods outperform self-supervised methods

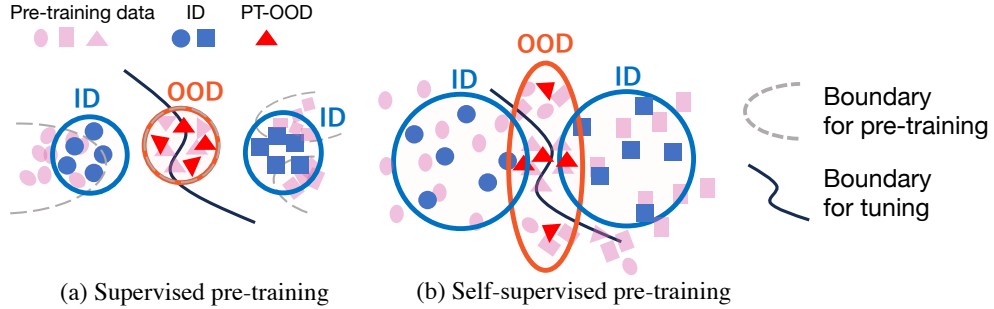

Figure 2: **Comparison of decision boundaries** for supervised and self-supervised pre-training.

for PT-OOD detection. Some work states that a strong ID classifier is necessary to detect unknown samples (Vaze et al., 2022). However, our results suggest that this is not a universal fact. We exhibit that a strong ID classifier is not always a robust OOD detector.

One might wonder if it is due to the poor performance of the simple OOD detector (Hendrycks & Gimpel, 2017) or the poor capability of the single linear layer. However, as shown in Table 2, this phenomenon occurs when the state-of-the-art OOD detectors, *e.g.,* Energy (Liu et al., 2020), ReAct (Sun et al., 2021), ReAct (Sun et al., 2021), MaxLogit (Hendrycks et al., 2022), Grad-Norm (Huang et al., 2021), and DICE (Sun & Li, 2022) are used or when replacing the last linear layer with MLP (3 layers). Note that we use post hoc (training-free) OOD detection methods (Yang et al., 2022) with ID decision boundaries to examine the robustness of the existing ID classifiers. Although only DICE, which prunes away noisy signals from unimportant feature units and weights, seems to achieve high performances, MoCo v2 still has a low performance. Therefore, we consider that this phenomenon does not result in a problem with the OOD detectors, but with the pre-training method.

**Reasons of robustness to PT-OOD.** We analyze that the reason for the high robustness of supervised methods is the higher linear separability on PT-OOD in the feature space. We show the overview of decision boundaries for both pre-training methods in Fig. 2. Since t-SNE (Van der Maaten & Hinton, 2008) or PCA (Wold et al., 1987) do not describe decision boundaries with the frozen pre-trained backbone, we show an illustration of each decision boundary. Supervised pre-training enforces the samples in the same class to have the same embeddings, so supervised pre-training allows the feature of PT-OOD to be linearly separable in the feature space. This is important for OOD detection because PT-OOD are not scattered but concentrated in the feature space, as shown in Fig. 2 (left). Therefore, they do not come inside the ID decision boundary, leading to high PT-OOD detection performance. On the other hand, self-supervised pre-training focuses on learning the transferable representations (Islam et al., 2021) and does not force the samples in the same classes to be embedded closely. Therefore, as shown in Fig. 2 (right), PT-OOD are scattered in the feature space and they might enter inside the ID decision boundary, resulting in low PT-OOD detection performance.

The transferability of pre-training algorithms is one of the strengths of self-supervised pre-training and is important to improve ID classification accuracy (Islam et al., 2021). Contrary to this, this study reveals that the high transferability may cause vulnerability in OOD detection.

## 5 APPLICATION TO RECENT FOUNDATION MODELS

Our findings also hold true when applied to recent large pre-trained models such as CLIP (Radford et al., 2021). It is true that existing CLIP-based OOD detection methods (Fort et al., 2021; Esmaeilpour et al., 2022; Ming et al., 2022a; Wang et al., 2023; Miyai et al., 2023b;a; Ming & Li, 2023) calculate the OOD scores by the similarity of image features and textual features, and do not use a linear projection. However, when applying CLIP to the domain where classes cannot be described by text, linear probing is a common approach for tuning CLIP. Since CLIP uses 400 million web-scraped data to learn transferable representations, we consider that most images in natural

Table 3: **Results for the OOD detection robustness of CLIP.** From this result, we find that CLIP also has a vulnerability to PT-OOD.

| ID | Backbone | Method | ID acc. (%) | AUROC (%) | | | | |
| | | | | IN-30 | IN-20 | iNaturalist | Places | SUN |
|---|---|---|---|---|---|---|---|---|
| CIFAR-10 | ResNet-50 | Sup. | **90.73** | **95.77** | **94.96** | **87.31** | **89.21** | **89.85** |
| | | CLIP | 87.92 | 61.06 | 63.66 | 68.65 | 68.46 | 65.92 |
| Caltech-101 | ResNet-50 | Sup. | **95.20** | - | **97.52** | **91.57** | **92.91** | **93.22** |
| | | CLIP | 93.60 | - | 86.37 | 73.23 | 79.87 | 80.69 |
| CIFAR-100 | ViT-B/16 | Sup. | 80.32 | **93.42** | **95.08** | **84.78** | **81.75** | **81.91** |
| | | CLIP | 82.39 | 68.04 | 69.90 | 71.66 | 61.41 | 61.09 |
| | | OpenCLIP | **83.05** | 73.54 | 79.04 | 80.06 | 73.34 | 73.05 |

photos can be PT-OOD. Therefore, we use ImageNet-30, ImageNet-20, iNaturalist (Van Horn et al., 2018), Places (Zhou et al., 2017), and SUN (Xiao et al., 2010) as PT-OOD. We use CLIP-ResNet-50 and CLIP-ViT-base/16 as a backbone. For CLIP-ViT-base/16, OpenCLIP (Ilharco et al., 2021), which is pre-trained with LAION-2B (Schuhmann et al., 2022), is also available, so we add experiments with OpenCLIP. Table 3 shows the results for OOD detection robustness with CLIP. From this result, we observe that CLIP has significantly low OOD detection robustness compared to the supervised pre-trained models. Since CLIP learns the transferable representations over various domains, PT-OOD can scatter in the feature space and enter within the ID decision boundaries, which is consistent with the analysis in Sec. 4. This result indicates that our analysis can be generalized to pre-trained models other than ImageNet-1K pre-training.

## 6    DETECTING PT-OOD WITH VARIOUS PRE-TRAINED MODELS

In this section, we explore the solution for detecting PT-OOD samples with various pre-training methods. As in previous experiments, existing approaches in OOD detection are to use ID decision boundaries to separate ID from OOD. However, when using large pre-trained models, this common approach is not the only option. Contrary to the common approach, when using pre-trained models, we can leverage the rich instance-by-instance discriminative representations of pre-trained models as they are to detect OOD, which does not utilize ID decision boundaries. Therefore, we aim to detect PT-OOD with the feature of pre-trained models. Note that existing feature-based detection methods (Sun et al., 2022; Lee et al., 2018) are intended to be applied to ID classifiers trained on ID data, so the usage to apply them directly to pre-trained models is non-trivial. For feature-based detection methods, we use kNN (Sun et al., 2022), which is a non-parametric density estimation using the nearest neighbor. The distance between a test input and its $k$-th nearest neighbor in the training ID data is used for OOD detection: $S_{\text{kNN}}(\mathbf{x}) = - \|\mathbf{z} - \mathbf{z}_k\|_2$, where $\mathbf{z}$ and $\mathbf{z}_k$ are the $L_2$ normalized embeddings, for the test input $\mathbf{x}$ and its $k$-th nearest neighbor. Although recent work (Uppaal et al., 2023) in the field of natural language processing has shown the effectiveness of applying kNN directly to pre-trained language models (*e.g.,* RoBERTa (Liu et al., 2021)), they did not focus on different pre-training algorithms and PT-OOD data, and also did not conduct experiments with vision pre-trained models. Hence, it alone cannot be generalized in our setting due to the difference in research domains.

Table 4 summarizes the comparison results with MSP and kNN. From this result, we find that in most settings, kNN (independent of the ID decision boundary) is more effective than MSP (dependent on the ID decision boundary). Especially when using MoCo, it can be seen that kNN is almost a perfect detector, even though MSP can be a random detector in some cases. From this result, it can be concluded that we can utilize the features of pre-trained models as they are to detect PT-OOD, without the use of ID decision boundaries.

Here, one might consider whether this approach would not be able to detect completely novel OOD data (non-PT-OOD) that networks do not know. As for a more versatile OOD detector that can detect PT-OOD and non-PT-OOD at the same time, based on our separate (PT-OOD/non-PT-OOD) discussions, we can take advantage of both methods with the ID decision boundaries for detecting

Table 4: **Results when applying kNN (Sun et al., 2022)** (feature-based detection methods) directly to pre-trained models. We use ImageNet-30 and ImageNet-20-O as OOD.

| OOD | IN-30 | | | IN-20 | | |
|---|---|---|---|---|---|---|
| ID | CIFAR-10 | CIFAR-100 | Food | CIFAR-10 | CIFAR-100 | Caltech-101 |
| **ResNet-50** | | | MSP / kNN | | | |
| Sup. | 95.77 / **99.99** | 79.98 / **100.00** | 93.57 / **99.94** | 94.96 / **100.00** | 81.42 / **100.00** | 97.52 / **97.93** |
| BYOL | 86.02 / **99.99** | 72.50 / **100.00** | 86.96 / **99.85** | 86.99 / **100.00** | 76.72 / **100.00** | 94.83 / **97.40** |
| SwAV | 81.05 / **100.00** | 71.67 / **100.00** | 83.12 / **99.86** | 83.94 / **100.00** | 71.72 / **100.00** | 92.12 / **96.21** |
| MoCo | 69.39 / **100.00** | 49.23 / **100.00** | 68.66 / **99.77** | 73.39 / **100.00** | 53.85 / **100.00** | 85.36 / **96.36** |
| **ViT-S** | | | | | | |
| Sup. | 98.16 / **100.00** | 89.54 / **100.00** | 88.78 / **99.81** | 97.76 / **100.00** | 90.78 / **100.00** | **98.61** / 98.56 |
| iBOT | 80.68 / **100.00** | 55.51 / **99.95** | 86.52 / **99.82** | 86.09 / **100.00** | 62.42 / **100.00** | 95.77 / **97.13** |
| DINO | 80.00 / **100.00** | 56.62 / **100.00** | 83.98 / **99.85** | 83.93 / **100.00** | 63.97 / **100.00** | 94.73 / **97.89** |

non-PT-OOD and methods without the ID decision boundaries for detecting PT-OOD. We show those results in Appendix B.3.

## 7 LIMITATIONS AND FUTURE WORK

**Extending to other visual recognition tasks.** In this work, we focus on only classification tasks because most studies in OOD detection address classification tasks (Yang et al., 2022). However, pre-trained models are widely used for other tasks. It is an interesting future direction to apply our findings to other tasks (Wang et al., 2021; Du et al., 2022b;a).

**Development of novel methods with ID decision boundaries.** Although we reveal that feature-based methods are effective for detecting PT-OOD, the drawback of the feature-based methods is an inference cost. Feature-based detection methods have to infer by querying whether or not it is close to the ID training data. Comparing methods with logits or probabilities for test data, feature-based methods have much inference cost. Therefore, it is future work to develop a robust and efficient OOD detection method for various pre-trained models.

**Application to other lightweight tuning methods.** In this study, we focus on linear probing because it is the most representative lightweight tuning method. These days, other lightweight tuning methods such as Adapter, which implements bottleneck adapter modules (Houlsby et al., 2019), visual prompt methods (Jia et al., 2022), LoRA (Hu et al., 2022), and their combinations (Chavan et al., 2023; Zhang et al., 2022) have been proposed for image classifications. We leave the application of our findings to these methods as important future work.

## 8 CONCLUSION

To examine the OOD detection robustness of current classifiers, this study focuses on a new question of whether pre-trained networks can accurately identify PT-OOD, whose information networks have. To answer this question, we explore the PT-OOD detection performance with different pre-training algorithms and indicate that the low linear separability of PT-OOD heavily degrades the PT-OOD detection performance. To solve this vulnerability, we further propose a solution to detect PT-OOD in the feature space independent of ID decision boundaries, which is unique to the use of pre-trained models.

**Broader Impact.** The main focus of this paper is to analyze the pre-training algorithm and PT-OOD detection performance. Although transferring pre-trained models is prevalent, this analysis has been largely overlooked in the literature. This analysis reveals the effectiveness of the state-of-the-art OOD detection methods depends on the pre-training algorithm and enables us to develop an effective way to use feature-based methods for pre-trained models. Considering the current situation where a number of detection methods have been proposed without using pre-trained models (Yang et al., 2022; Zhang et al., 2023), we believe that the analysis in this paper has a solid contribution to OOD detection with pre-trained models even if we do not propose a completely novel method.

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

APPENDIX

This appendix provides an illustration of our problem setting (Appendix A), additional experimental results (Appendix B), implementation details (Appendix C), and our source code (Appendix D).

## A ILLUSTRATION OF OUR PROBLEM SETTING

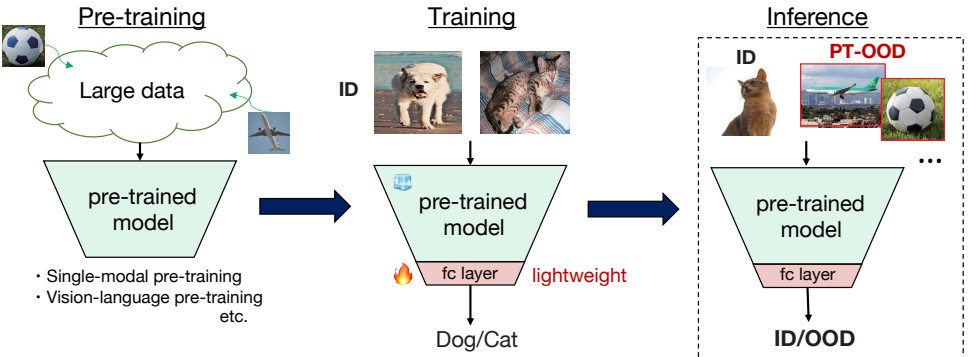

Figure 3: Illustration of the experimental setting of PT-OOD detection.

In Fig. 3, we illustrate our problem setting for PT-OOD detection. During test time, we introduce a new OOD data called PT-OOD, which contains samples similar to those seen during pre-training. The model is initially pre-trained with a large-scale dataset and then trained with linear probing with ID data. During test time, we detect PT-OOD. The critical point is that the backbone network has information on PT-OOD. This setting is crucial to ensure the safety of current classifiers with large pre-trained knowledge.

## B ADDITIONAL EXPERIMENTS

### B.1 RESULTS OF FEATURE-BASED METHODS WITH CLIP

Table 5 shows the results of feature-based OOD methods with CLIP. We can see that kNN is also more effective than MSP for CLIP.

### B.2 HISTOGRAM OF SCORES WITH ID AND PT-OOD

To investigate the distribution of confidence of ID and PT-OOD, we show the histogram of the ID classifier's MSP with ID and PT-OOD in Fig. 4. Self-supervised methods confuse ID and PT-OOD samples, while the supervised method can clearly separate ID and PT-OOD samples. In particular, the confidences for PT-OOD are different, although the confidences for ID are almost the same, which means that self-supervised pre-trained models tend to misclassify PT-OOD samples as ID samples.

Table 5: **Results for feature-based methods with CLIP**. We compare MSP and kNN.

| ID | CIFAR-10 | Caltech-101 | CIFAR-100 |
|---|---|---|---|
| OOD | IN-30 | IN-20 | IN-30 |
| | | MSP / kNN | |
| CLIP | 61.06 / **98.87** | 86.37 / **90.36** | 68.04 / **99.97** |

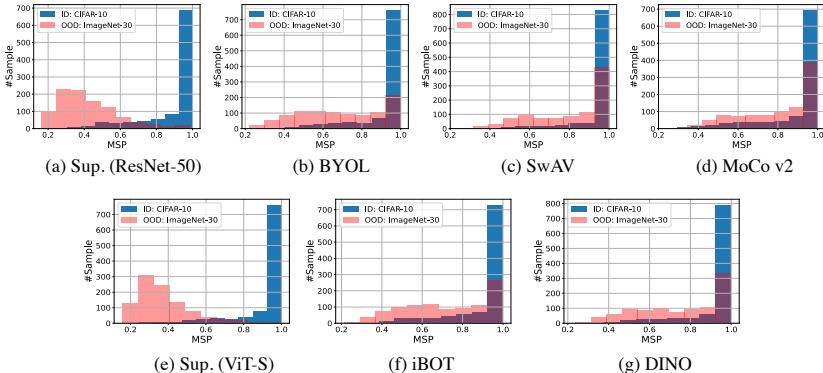

Figure 4: **Histogram of the confidences with MSP** on ID (CIFAR-10) and PT-OOD (ImageNet-30). Self-supervised pre-trained models produce incorrect high confidence to PT-OOD, while supervised pre-trained methods ensure separation between ID and PT-OOD.

Table 6: **Results for OOD detection on both non-PT-OOD and PT-OOD.** We use CIFAR-10 (ID), CIFAR-100 (non-PT-OOD) and ImageNet-30 (PT-OOD).

| Method | Source | Sup. C100 | Sup. IN-30 | BYOL C100 | BYOL IN-30 | SwAV C100 | SwAV IN-30 | MoCo v2 C100 | MoCo v2 IN-30 | AVG |
|---|---|---|---|---|---|---|---|---|---|---|
| MSP | prob | 88.31 | 95.77 | 86.64 | 86.02 | 84.98 | 81.05 | 84.84 | 69.39 | 84.63 |
| Energy | logit | 90.48 | 98.90 | 90.00 | 91.80 | 87.73 | 84.88 | 85.63 | 62.16 | 86.45 |
| MaxLogit | logit | 90.64 | 98.60 | 90.16 | 91.81 | 87.86 | 84.99 | 86.01 | 62.95 | 86.63 |
| KL Matching | prob | 80.99 | 93.50 | 80.66 | 82.93 | 76.52 | 78.32 | 78.28 | 73.49 | 80.59 |
| GradNorm | grad norm | 79.87 | 93.31 | 77.29 | 69.14 | 73.99 | 69.44 | 75.27 | 31.81 | 71.27 |
| ODIN | prob+grad | **91.18** | 99.44 | 90.47 | 98.43 | 88.60 | 97.96 | 87.00 | 86.99 | 92.50 |
| ReAct | logit | 91.06 | 99.16 | **91.09** | 92.71 | 88.49 | 86.97 | 81.79 | 66.31 | 87.20 |
| DICE | logit+pruning | 85.46 | 99.74 | 81.16 | 97.32 | 76.52 | 98.60 | 76.90 | 86.80 | 87.81 |
| Residual | **feat** | 76.44 | **100.00** | 73.42 | 99.97 | 73.22 | **100.00** | 73.01 | **100.00** | 87.01 |
| Mahalanobis | **feat**+label | 76.55 | **100.00** | 72.81 | **100.00** | 73.14 | **100.00** | 73.02 | **100.00** | 86.94 |
| kNN | **feat** | 81.55 | **99.99** | 83.83 | **99.99** | 80.46 | **100.00** | 76.97 | **100.00** | 90.34 |
| ViM | **feat**+logit | 89.90 | **100.00** | 90.45 | **100.00** | 90.75 | **100.00** | 89.64 | **100.00** | **95.09** |

## B.3 VERSATILE OOD DETECTORS FOR DETECTING PT-OOD AND NON-PT-OOD

In this section, we explore the versatile OOD detector for detecting PT-OOD and non-PT-OOD with various pre-training methods. In the main experiments, we focus on PT-OOD detection since exploring PT-OOD detection is crucial to bridge the gap between the practice of OOD detection and current classifiers. However, in the real world, it is unknown whether PT-OOD or non-PT-OOD comes as OOD data, and non-PT-OOD samples can not always be identified only with the feature embeddings because pre-trained models might not have information. Therefore, we explore versatile OOD methods that can detect both PT-OOD and non-PT-OOD.

To explore the versatile OOD detectors, our separate discussion about non-PT-OOD and PT-OOD is helpful. In detail, for detecting non-PT-OOD, Table 1 shows that the methods with the ID decision boundary perform well. For detecting PT-OOD, Table 4 shows that the feature-based methods with pre-trained models are effective. As a result of this separate analysis, we can hypothesize that methods that ensemble both ID decision boundary-based and feature-based methods are versatile OOD detectors. In Table 6, we show the results of non-PT-OOD and PT-OOD detection with many state-of-the-art OOD detection methods. We use CIFAR-10 as ID, CIFAR-100 as non-PT-OOD, and ImageNet-30 as PT-OOD. That is true that the methods such as kNN (Sun et al., 2022), Mahalanobis (Lee et al., 2018), and Residual (Wang et al., 2022) can detect PT-OOD samples almost perfectly, but the non-PT-OOD detection performance is lower than some probability-based or logit-based methods. On the other hand, although some probability-based or logit-based methods, such as Maxlogit (Hendrycks et al., 2022) or ReAct (Sun et al., 2021), can detect non-PT-OOD samples,

Table 7: **Results for full fine-tuning.** We use MSP as an OOD detection method. We use CIFAR-10 as the ID data. Although supervised pre-trained models are more robust, the difference in AUROC between pre-training methods becomes small.

| Backbone | Method | #Params | IN-30 |
|---|---|---|---|
| ResNet-50 | Sup.† | 23M | **97.09** |
| | BYOL | 23M | 95.65 |
| | SwAV | 23M | 94.18 |
| | MoCo v2 | 23M | 93.04 |
| ViT-S | Sup.† | 22M | **99.43** |
| | DINO | 22M | 99.13 |

Table 8: **Results on OOD data with only visual similarity.** We use MSP as a detection method. We use CIFAR-10 as ID and ImageNet-30 (IN-30), iNaturalist (Van Horn et al., 2018), and SUN (Xiao et al., 2010) as OOD.

| Method | IN-30 | iNaturalist | SUN |
|---|---|---|---|
| Sup. | **95.77** | 87.31 | **89.75** |
| BYOL | 86.02↓-9.75 | 83.63↓-3.68 | 88.40↓-1.35 |
| SwAV | 81.05↓-14.72 | **87.45**↑+0.14 | 88.69↓-1.06 |
| MoCo v2 | 69.39↓-26.38 | 73.79↓-13.52 | 78.69↓-11.06 |

they all fail to detect PT-OOD samples. The method with the highest performance for both PT-OOD and non-PT-OOD detection is ViM (Wang et al., 2022), which combines the class-agnostic score from feature space and the ID class-dependent logits. This result is consistent with our hypothesis above and thus demonstrates the benefit of the separate analysis in the main paper.

### B.4 RESULTS WITH FULL FINE-TUNING

We investigate the OOD detection performance with full fine-tuning, which updates all the parameters. In the main experiments, we used linear probing because we focused on light-weight tuning. However, it is also unclear from existing research how full fine-tuning affects PT-OOD detection performance. In Table 7, we show the experimental results for full fine-tuning. These results show that, although supervised pre-trained models are more robust, the difference in AUROC between pre-training methods becomes small. This is because the information about PT-OOD data in the backbone has been lost by full tuning (*i.e.*, catastrophic forgetting (Goodfellow et al., 2013)). However, considering that transferring pre-trained models is prevalent, our findings with linear probing are crucial to ensure the safety of current ID classifiers.

### B.5 RESULTS WITH OOD DATA WITH ONLY VISUAL SIMILARITY

In previous experiments, we used iSUN/LSUN/CIFAR-100 datasets as non-PT-OOD because previous studies used such datasets as OOD (Koner et al., 2021). However, the images in these OOD datasets are low-resolution, and the samples in these datasets are not visually similar to the ImageNet samples. Therefore, the question remains about how samples, that do not have ImageNet classes but have high visual similarity, will behave. In this experiment, we use iNaturalist (Van Horn et al., 2018) and SUN (Xiao et al., 2010) as OOD data. These datasets are provided without ImageNet-1K classes (Huang et al., 2021), but the samples in these datasets have high visual similarity to ImageNet. In Table 8, we show the results on these data with MSP and Mahalanobis. For MSP, the results demonstrate that although MoCo v2 still has a lower detection performance, SwAV and BYOL are comparable to the supervised method. The reason is that the difference in representation ability between the supervised learning method and self-supervised learning methods has been reduced because of semantics differences from ImageNet. However, as shown in the experiments with CLIP (Table 3), when pre-trained on larger amounts of data, the boundary between PT-OOD and non-PT-OOD becomes unclear and the number of PT-OOD will increase.

## C  IMPLEMENTATION DETAILS

### C.1  ID DATASET

**CIFAR-10 and CIFAR-100.** Following the existing literature (Liang et al., 2018; Koner et al., 2021; Fort et al., 2021; Ming et al., 2022b), we use CIFAR-10 (Krizhevsky, 2009) and CIFAR-100 (Krizhevsky, 2009) as ID datasets. For CIFAR-10 and CIFAR-100, the training and validation partition of the dataset and the evaluation data follow existing work (Koner et al., 2021; Fort et al., 2021). We randomly sample 1,000 images at test time to balance the number of ID and OOD data. The CIFAR-10 and CIFAR-100 samples are provided in a size of 32×32, but we used 224×224 following the existing study (Koner et al., 2021) to better utilize the knowledge of the pre-trained models.

**Caltech-101.** Caltech-101 consists of pictures of objects belonging to 101 classes and has about 40 to 800 images per category. For Caltech-101, we exclude "Faces easy" and "Faces" classes due to the overlap of OOD data, and use 99 classes for training. We use 1,000 samples for the test and the remaining for training. We use an image size of 224×224.

**Food-101.** Food-101 is a real-world food dataset containing the 101 most popular and consistently named dishes. Food-101 consists of 750 images per class for training and 250 images per class for testing. We use the official train and validation partition of the dataset. We use an image size of 224×224.

### C.2  PT-OOD DATASET

**ImageNet-30.** We use ImageNet-30 test data (IN-30) (Hendrycks et al., 2019b), which consists of 30 classes selected from validation data and an expanded test data of ImageNet-1K. To avoid the overlap of the semantics of images with ID data, we exclude some classes and samples from ImageNet-30. When the ID dataset is CIFAR-10, following the existing study (Tack et al., 2020), we exclude "airliner", "ambulance", "parkingmeter", and "schooner" classes. When the ID dataset is CIFAR-100, we use "acorn", "airliner", "american alligator", "digital clock", "dragonfly", "hotdog", "hourglass", "manhole cover", "nail", "revolver", "schooner", "strawberry", "toaster" as OOD classes. In addition, we exclude samples in which humans highly likely appear because CIFAR-100 has labels related to humans (*i.e.,*, woman, man, child). When the ID dataset is Food, we exclude "hotdog" class. We randomly sample 1,000 images to balance the number of ID and OOD data.

**ImageNet-20-O.** We create ImageNet-20-O, which consists of 20 classes selected from the validation data of ImageNet-1K. To avoid the overlap of the semantics of images with ID data, we use n01914609, n02782093, n03047690, n03445777, n03467068, n03530642, n03544143, n03602883, n03733281, n03804744, n03814906, n03908714, n03924679, n03944341, n04033995, n04116512, n04398044, n04596742, n07716358, and n12768682 from ImageNet-1K. We also manually exclude some samples that have similar semantics to those in ID datasets. ImageNet-20-O has 940 samples, and we use all samples for testing.

### C.3  NON-PT-OOD DATASET

We mainly use LSUN, iSUN, CIFAR-100 (when ID is CIFAR-10), and CIFAR-10 (when ID is CIFAR-100) datasets as non-PT-OOD. These datasets are provided not to overlap with CIFAR-10/CIFAR-100 classes. In addition, these datasets are created by downsampling original images to 32×32 and then upsampling to 224×224 when applied to pre-trained models (Koner et al., 2021). We randomly sample 1,000 images to balance the number of ID and OOD data.

### C.4  SELF-SUPERVIESD PRE-TRAINED MODELS

For SwAV and MoCo v2, we use the model pre-trained for 200 epochs, which are officially provided. For BYOL, the models pre-trained for 200 epochs are not officially provided. Therefore, we used the model pre-trained for 300 epochs provided by https://github.com/yaox12/BYOL-PyTorch, whose reproduction result is very similar to that of the original paper. For iBOT

and DINO, we used the model pre-trained for 800 epochs because it is the only model officially provided for ViT-S.

## C.5    TRAINING SETUP

For augmentations, we use RandomHorizontalFlip and Resize for training and Resize for testing. We normalize images with ImageNet statistics. As an optimizer, we use SGD for ResNet and AdamW for ViT. The learning rate of each ID classifier is tuned to achieve high ID accuracy.

## D    SOURCE CODE

The source code we used in the experiments is provided in the supplementary. Please refer to the README file in the code directory for details.

