# OpenReview forum: "Can Pre-trained Networks Detect Familiar Out-of-Distribution Data?"
_ICLR.cc/2024/Conference — ICLR 2024 Conference Withdrawn Submission_

### Official Review · Reviewer_cmE8 · 2023-10-27

**Soundness:** 2 fair
**Presentation:** 3 good
**Contribution:** 2 fair
**Rating:** 3
**Confidence:** 4

**Summary:**

This work presents an empirical analysis of the detection performance of existing detectors and pre-training algorithms on a kind
of out-of-distribution data entitled as pre-trained OOD (PT-OOD). PT-OOD are samples from the distribution used to pre-train
the model that is different from the distribution used to finetune the model.
They run experiments with supervised and self-supervised
learning algorithms to pre-train a backbone on a large dataset (ImageNet-1K) and finetune it with a linear layer head (linear probe)
with the backbone frozen on a smaller dataset. They found out that, even though feature-based methods achieve stellar
performance distinguishing PT-OOD data from in-distribution data (ID), methods based on the outputs of the finetuned model
suffer from detecting PT-OOD.

**Strengths:**

They tackle an original question of whether the source data used for pre-training a model that does not overlap with the
target data the model was adapted to can be easily detected by showcasing empirical results with a couple of backbone architectures
and a few learning algorithms.

To the best of my knowledge, they are the first to uncover that using simply the features of a model
with a simple method, such as kNN density estimation, one can almost perfectly distinguish PT-OOD from ID data. They also showcase
limitations on the MSP for such a task and other detectors that rely on the logits or the decision boundaries between ID classes
to make a prediction.

**Weaknesses:**

This paper tries to motivate the problem with web-scale models but conducts supervised pre-training on ImageNet-1K only. When actually using web-scale datasets, supervised pre-training becomes unfeasible.

This does not invalidate the results found but hinders the clarity and true contribution of the manuscript.
Experiments on Dino v2 trained on a web-scale dataset would be appreciated, for instance. The model weights are easily available online.

Figure 2 is a conceptual drawing. It would be nice to see a data-driven approach to support the claims of the authors
by, for instance, comparing the decision boundaries of a linear probe for supervised and self-supervised pre-training on a
binary classification task.

I disagree with the claim on page 3:

> since a large amount of pre-training data is scraped from the web, it is more likely for PT-OOD to come as input.

I reckon that the standard use case for classification-as-a-service applications is users inputting their own originally acquired data into the system, not necessarily scraped from the web.

Some claims are too strong or are unsupported by sufficient evidence throughout the paper.

**Questions:**

1. What can be changed on logits/softmax based approaches to improve their performance on PT-OOD detection? The authors explores
mainly methods that do not have a reference from the ID dsitribution to perform density estimation similar to kNN. I suggest authors
try to perform experiemnts with kNN on the logits and investigate methods such as KL-Matching [2] or Igeood [3] that compares test samples to training prototypes.
2. Would the same conclusions be observed by running on self-supervised experiments (e.g., DinoV2 [1]) trained on a web-scale dataset?
3. Does a data-driven approach of Figure 2 show the same as what is conceptualized by the authors?

I believe that suggesting improvements for logits/softmax-based methods (Question 1) or explaining their failure by backing the authors'
intuition with data (Question 3) could improve the paper and change my opinion favorably.

References:

[1] Oquab et al. "DINOv2: Learning Robust Visual Features Without Supervision." CVPR 2023. /abs/2304.07193.\
[2] Hendrycks et al. "Scaling Out-of-Distribution Detection for Real-World Settings." ICML 2022. /abs/1911.11132\
[3] Gomes et al. "Igeood: An Information Geometry Approach to Out-of-Distribution Detection." ICLR 2022. /abs/2203.07798

---

### Official Review · Reviewer_PtvE · 2023-10-29

**Soundness:** 1 poor
**Presentation:** 2 fair
**Contribution:** 3 good
**Rating:** 3
**Confidence:** 3

**Summary:**

This paper proposes to study the OOD sample detection for pre-trained models where some OOD data may be in the pretraining dataset (PT-OOD). It is observed that the detection performance for self-supervised pretrained model is worse than supervised pretrained model. The paper propose to use k-NN to detect PT-OOD sample.

**Strengths:**

The problem setting of detecting PT-OOD samples is interesting and might be valuable for future application of large pre-trained models.

**Weaknesses:**

1). The analysis lacks support. The paper report that OOD detection methods perform better on supervised pretrained model than on self-supervised pretrained model. The analysis in this paper says that it is because the features of models under supervised pretraining are linear seperable while the features of self-supervised trained models are not. Except the illustration in Fig.2, I have not found any theoretical or empirical evidence to support this analysis.

2). The proposed method lacks novelty and contradicts to the analysis in section 4. This paper proposes to apply kNN on features to detect PT-OOD samples. While applying clustering method on features to classify data sample is a conventional way in feature laerning [1], the proposed method lacks novelty. Furthermore, it contradicts to the analysis in section 4, stating the PT-OOD features scatter among ID features.

3). Some part of the paper is confusing. For example, in the last paragraph in the introduction, it first says " We can utilize instance-by-instance discriminative features to separate ID and OOD, which require ID boundaries" and in the next phrase, it says "without using ID decision boundaries", which is confusing.

[1] Bingyi Kang, Saining Xie, Marcus Rohrbach, Zhicheng Yan, Albert Gordo, Jiashi Feng, and Yannis Kalantidis. Decoupling representation and classifier for long-tailed recognition. In ICLR 2020.

**Questions:**

I have major concerns over the analysis in this paper.

1). Why whould the features from self-supervised model be less linear seperable than supervised model? By tuning the last linear layer, self-supervised models achieve similar or better performance than supervised models, it is not obvious why less linear seperable is the reason for less robustness in detecting PT-OOD samples.

2). If the analysis in this paper is true, why would kNN be an effective method to detect PT-OOD sample when "PT-OOD can scatter in the feature space"?

Therefore, I think the paper requires a more in-depth analysis of the PT-OOD problem.

---

### Official Review · Reviewer_rG3x · 2023-11-01

**Soundness:** 2 fair
**Presentation:** 2 fair
**Contribution:** 2 fair
**Rating:** 5
**Confidence:** 5

**Summary:**

The paper analyzes the impact of pretraining for OOD detection. Its main contributions are (1) Supervised pretraining is better than self-supervised pretraining for the downstream OOD detection task. (2) kNN is more effective than maximum softmax probability for OOD detection when the model is a pretrained one.

**Strengths:**

1. On CIFAR and ImageNet-1k datasets, the experiment scope is extensive.
2. The paper is clearly written.

**Weaknesses:**

1. The impact of post-processors (the scoring function) is huge, but the authors did not explore the state-of-the-art scoring functions such as ViM [1] and NNGuide [2]
2. It is not clear what exactly is meant by 'instance-by-instance' discriminative representation.
3. The experiments with models pretrained on ImageNet-21/22K are not available.
4. One of the main observations (i.e., supervised > self-supervised) is too similar to the observation noted in [3]

[1] Wang, Haoqi, et al. "Vim: Out-of-distribution with virtual-logit matching." Proceedings of the IEEE/CVF conference on computer vision and pattern recognition. 2022.
[2] Park, Jaewoo, Yoon Gyo Jung, and Andrew Beng Jin Teoh. "Nearest Neighbor Guidance for Out-of-Distribution Detection." Proceedings of the IEEE/CVF International Conference on Computer Vision. 2023.
[3] How to Exploit Hyperspherical Embeddings for Out-of-Distribution Detection?

**Questions:**

Please address the weaknesses mentioned in the above

---

### Official Review · Reviewer_wQPP · 2023-11-02

**Soundness:** 3 good
**Presentation:** 3 good
**Contribution:** 2 fair
**Rating:** 5
**Confidence:** 3

**Summary:**

This work considers a class of OOD samples elicited by the use of pre-trained models in OOD Detection: pre-Trained OOD, referring to OOD samples memorized by pre-trained models. The authors explored the performance of PT-OOD detection under different pre-training algorithms through a number of experiments, and the results showed that the low linear separability of PT-OOD severely degraded the PT-OOD detection performance. They further proposed a solution based on a large-scale pre-trained feature extractor.

**Strengths:**

1. The PT-OOD detection problem induced by the use of pre-trained models is interesting.
2. Extensive experimental analyses were conducted.
3. The influence of self-supervised and supervised pre-training are investigated.

**Weaknesses:**

1. The definition of PT-OOD is very ambiguous.
2. The article gives a weak motivation for the study and fails to see why targeted testing for PT-OOD is important.
3.  In the experiments, the pre-training is not large-scale.

**Questions:**

Q1. Dose the downstream task model have very poor prediction performance on PT-OOD samples? What is the importance of targeting PT-OOD as opposed to detecting OOD samples?

Q2. What is the relationship between pre-training data and ID data for downstream detection tasks? When doing PT-OOD detection, is the groundtruth result AUROC=1 or FPR=0?